# ER Stress Response and Induction of Apoptosis in Malignant Pleural Mesothelioma: The Achilles Heel Targeted by the Anticancer Ruthenium Drug BOLD-100

**DOI:** 10.3390/cancers14174126

**Published:** 2022-08-26

**Authors:** Elia Ranzato, Gregorio Bonsignore, Simona Martinotti

**Affiliations:** DiSIT-Dipartimento di Scienze e Innovazione Tecnologica, University of Piemonte Orientale, Viale Teresa Michel 11, 15121 Alessandria, Italy

**Keywords:** apoptosis, calcium, endoplasmic reticulum, GRP78, malignant pleural mesothelioma, mitochondria, ROS, unfolded protein response (UPR)

## Abstract

**Simple Summary:**

Malignant mesothelioma is a rare cancer arising from the serosal surfaces of the body, mainly from the pleural layer. This cancer, strongly linked to asbestos exposure, shows a very inauspicious prognosis. In fact, there is no efficient therapeutic treatment for malignant pleural mesothelioma (MPM). Thus, there is an urgent need to develop novel therapeutic approaches to treat this form of cancer. Our previous study showed the importance of GRP78 in MPM survival. BOLD-100 is a specific modulator of GRP78 and we have observed that it shows cytotoxicity against MPM cells. In particular, we describe that BOLD-100 increases oxidative stress and deregulates the calcium homeostasis leading to cell stress and, ultimately, to cell death. Our in vitro data strongly suggest that BOLD-100 inhibits the growth of MPM cell lines, proposing the application as a single agent, or in combination with other standard-of-care drugs, to treat MPM.

**Abstract:**

Malignant mesothelioma is a rare cancer arising from the serosal surfaces of the body, mainly from the pleural layer. This cancer is strongly related to asbestos exposure and shows a very inauspicious prognosis, because there are scarce therapeutic options for this rare disease. Thus, there is an urgent need to develop novel therapeutic approaches to treat this form of cancer. To explore the biology of malignant pleural mesothelioma (MPM), we previously observed that MPM cell lines show high expression of the GRP78 protein, which is a chaperone protein and the master regulator of the unfolded protein response (UPR) that resides in the endoplasmic reticulum (ER). Based on our previous studies showing the importance of GRP78 in MPM, we observed that BOLD-100, a specific modulator of GRP78 and the UPR, shows cytotoxicity against MPM cells. Our studies demonstrated that BOLD-100 increases ROS production and Ca^2+^ release from the ER, leading to ER stress activation and, ultimately, to cell death. Our in vitro data strongly suggest that BOLD-100 inhibits the growth of MPM cell lines, proposing the application as a single agent, or in combination with other standard-of-care drugs, to treat MPM.

## 1. Introduction

Malignant mesothelioma is a rare malignancy of mesothelial cells that can manifest at any mesothelial layer, but the pleural layer is the most frequently affected, leading to the development of malignant pleural mesothelioma (MPM).

Currently, MPM incidence is significantly lower than that of new lung cancer cases, but it has gradually grown in the past five decades [1]. The World Health Organization (WHO) assesses that in the world every year, more than 100,000 deaths are asbestos-related (i.e., asbestosis, lung cancer, and mesothelioma). In the 2003–2014 interval, in Italy 16,086 persons died from MPM, corresponding to 1340 per year [2]. U.S. mesothelioma statistics from 1999 to 2016 account for 44,538 men and 13,119 women diagnosed with MPM. Mortality data for MPM are not available for Russia, China, or India, according to the WHO [3].

The onset of MPM is related to the widespread use of asbestos as an insulator since the 1950s in fields such as construction, pipefitting, shipbuilding, and car brake assembly due to its low cost, light weight, and high heat resistance. Asbestos is a natural silicate mineral that can be categorized as amphibole, which is characterized by linear and needle-like fibers, and as a serpentine structure composed of spiral fibers.

Typically, MPM spread follows continued asbestos exposure of two to three decades [4]. Histologically MPM can be classified in two variants, namely epithelioid and sarcomatoid subtypes, but some tumors display characteristics of both types and are thus classified as biphasic [5]. Treatment strategies are dependent on patient characteristics, and disease phase at diagnosis. Unfortunately, due to the long latency period, most patients show first symptoms only with advanced disease. Chemotherapy is the suggested approach, although life expectancy for advanced disease is very short [6]. Very recently, in Brazil and the U.S., immunotherapy (i.e., nivolumab plus ipilimumab), is now indicated as a first-line treatment for unresectable mesothelioma [7,8]. However, there is an urgent need for new and more distinct targeted therapies to improve patient outcomes.

The ER chaperone protein 78-kDa glucose-regulated protein (GRP78) is often elevated in a broad range of human cancers where it activates pro-survival pathways promoting tumor survival and proliferation, and confers resistance to routinely utilized chemotherapeutic agents (i.e., cisplatin, paclitaxel, doxorubicin), making it an attractive cancer target [9,10,11,12,13].

Cancer cells are characterized by an altered glucose metabolism, while the tumor microenvironment is marked by impaired blood flow and hypoxia. Under such conditions, tumor cells overexpress GRP78 that promotes the UPR, allowing cell survival by blocking pro-apoptotic functions. In addition, GRP78 relocates to the plasma membrane where it acts as a co-receptor, regulating MAPK and PI3K/AKT pathways [14,15]. Such a dual behavior of GRP78 is thought to be at the core of developing chemoresistance [16]. Various studies have found an association between GRP78 overexpression and negative patient outcome, suggesting that targeting the function of the UPR may augment the efficacy of chemotherapeutic regimens, especially in recalcitrant tumors [17]. The delicate equilibrium between pro-survival and death processes determined in cancer cells by the constitutive UPR suggests the possibility of targeting GRP78 as a cancer weak point.

Our data concerning increased basal GRP78 expression levels in MPM cells are in agreement with previous immunohistochemical data on MPM biopsy tissues [18], overall indicating that a mild UPR is a constitutive MPM condition. Such a view is further corroborated by our finding of GRP78 expression at the cell surface, and in addition by the observed positive correlation between cell surface GRP78 levels and the degree of AKT phosphorylation [19].

BOLD-100 (sodium trans-[tetrachlorobis(1H-indazole) ruthenate(III)]) is a clinical-stage therapeutic that has previously been tested in a phase 1 clinical trial as a monotherapy in advanced cancers [20], and is currently being tested in a phase 1b/2 clinical trial in combination with FOLFOX (fluorouracil, leucovorin and oxaliplatin) for the treatment of advanced gastrointestinal cancers (NCT04421820). BOLD-100 has a complex mechanism of action in cancer cells, with one of the major pathways being the prevention of the GRP78 upregulation, downstream activation of the UPR pathway, and subsequent cell death through caspase-8 activation [21], with minimal impact on normal human cell lines and primary cells [12].

Based on our previous data demonstrating the upregulation of GRP78 in MPM cell lines [19], the aim of this current study was to evaluate the mechanistic effect of the clinical-stage small molecule drug BOLD-100 on the MPM cells in inducing cellular toxicity.

## 2. Materials and Methods

### 2.1. GRP78 Inhibitor

The inhibitor of GRP78 (BOLD-100, sodium trans-[tetrachlorobis(1H-indazole) ruthenate(III), with cesium as an intermediate salt form], previously known as KP-1339, NKP1339 and IT-139) was provided by Bold Therapeutics Inc. (Vancouver, Canada).

### 2.2. Cell Culture and Reagents

Unless specified, all reagents were purchased from Merck (Milan, Italy).

For the experiments, we utilized the following human MPM cell lines: REN, a p53-mutant, inflammatory epithelial subtype cell line [22]; MM98 cells, established from the pleural effusion of a sarcomatous MPM [23]; Met5A, a non-cancerous human mesothelial cell line, purchased from ATCC [24]. Cells were cultured in a humidified atmosphere (5% CO_2_) at 37 °C, by using complete medium (DMEM (high glucose, 4.5 g/L), 10% foetal bovine serum (FBS, Euroclone, Pero, Italy), 100 U/mL penicillin, 100 mg/mL streptomycin) and L-glutamine (200 mM) [25].

### 2.3. Cytotoxicity Assay

The evaluation of BOLD-100 cytotoxicity was carried out by using calcein-AM (calcein acetoxymethyl ester) assay [26]. Cells growing in 96-well plates were treated for 24 or 48 h with an increasing concentration of the drug, washed with PBS before incubation with 2.5 μM calcein-AM in PBS, at 37 °C for 30 min. The resulting fluorescence was read (485-nm excitation and 535-nm emission) with the Infinite 200 Pro plate reader (Tecan, Vienna, Austria).

### 2.4. Colony-Forming Assay

REN cells (500–1000) were plated in six-well dishes and incubated at 37 °C, 5% CO_2_ for colony formation. After 8 days, colonies were fixed with 3.7% paraformaldehyde (PFA) for 15 min and stained with 0.5% crystal violet for 10 min for colony visualization. After crystal violet staining, colonies containing more than 50 cells were counted and evaluated [27].

### 2.5. Activity of Caspase 3 and Caspase 8

Cells seeded in a 96-well plate were treated with 100 µM BOLD-100 in the presence or absence of a specific caspase 3/7 inhibitor (N-Ac-DEVD-CHO) and incubated at 37 °C before measurement of caspase 3/7 activity, as indicated in the manufacturer’s protocol (Cayman Chemicals).

The activity of caspase-8 was assessed with the Caspase-8 Detection Kit (FITC-IETD-FMK) according to the manufacturer’s instruction. For the inhibitory experiment, cells were treated with the specific inhibitor of caspase-8 (Z-IETD-FMK).

### 2.6. Calpain Activity

REN cells were seeded overnight in 96-wells plate, then incubated at 37 °C for 30 min with 20 μM t-BOC (t-butoxycarbonyl)-Leu-Met-CMAC(7-amino-4-chlorometylcoumarin) fluorogenic calpain substrate. After incubation, cells were washed twice, and the fluorescence emission was monitored with a plate reader (ex: 355 nm/em: 485 nm) [28].

Calpain activity was determined in the absence or presence of 100 μM BOLD-100, and in the presence of a calpain inhibitor (PD150606).

### 2.7. Mitochondrial Permeability Transition Pore Assay

To examine the effect of the BOLD-100 on mitochondrial membrane integrity loss, calcein-AM cobalt (CoCl_2_) assay was performed. Cells seeded on a 96-well plate were loaded with calcein-AM, which diffuses into the cells passively and accumulates in the mitochondria and the cytosol to liberate the highly polar fluorescent dye calcein. CoCl_2_ (cobalt chloride, 1 mM) quenches the cytosolic fluorescence, while mitochondrial fluorescence is maintained. The fluorescence change was assessed using a plate reader (ex 485 nm/em 535 nm).

### 2.8. JC-1 Fluorescence/JC-1 Mitochondrial Membrane Potential Assay

Bulk red and green fluorescence from cells labeled with JC-1 [29] were measured with the Infinite 200 Pro well plate reader (Tecan, Vienna, Austria) in 96-well plates. Red (ex = 535, em = 590 nm) and green (ex = 485, em = 535) fluorescence values were acquired, and red/green ratios were calculated to indicate the mitochondrial membrane potential.

### 2.9. Measurements of free Cytosolic Ca^2+^ Concentration ([Ca^2+^]_i_)

For the measurement of free cytosolic Ca^2+^, MPM cells were cultured on glass-base dishes (Iwaki Glass, Inc., Tokyo, Japan). After overnight attachment, cells were loaded (loading buffer as described in a previous work [25]) in the dark with 20 µM Fluo-3/AM (a fluorescent, cell-permeant calcium probe) at 37 °C for 30 min. After probe loading and washing, confocal imaging was performed by time-lapse analysis (resolution: 512 × 512 pixels at 256 intensity values, framing rate: 1 frame/5 s, excitation: 488 nm Ar laser (laser power was reduced to 0.5% in order to lower probe bleaching), emission: 505–550 bandpass filter), using a Zeiss LSM 510 confocal system interfaced with a Zeiss Axiovert 100 M microscope (Carl Zeiss Inc., Oberkochen, Germany). Fluorescence from several cells was acquired using a 20× objective, utilizing ROI mean tool software. Probe calibration was realized utilizing the Grynkiewicz equation [30]:Ca^2+^ = Kd(F − Fmin)/(Fmax − F)
where Kd = 400 nmol/L. Fmin and Fmax are minimum and maximum fluorescence intensities achieved by Fluo-3/AM calibration after the exposure of cells to 500 μM A23187 for about 10 min, followed by 20 mM EDTA for 2 min.

### 2.10. Measurements of Intracellular Reactive Oxygen Species (ROS)

The fluorescent dye precursor dihydrorhodamine (DHR)-123, which is converted by ROS into the fluorescent analogue rhodamine-123, was used to evaluate their intracellular level. 

Cells were plated in a 96-well plate, allowed to attach overnight, and then loaded with DHR-123 (30 µM) in a loading buffer, at room temperature in the dark over 30 min [31].

DHR 123 fluorescence was read with a plate-reader by using a 485 nm excitation filter and a 530 nm emission filter. ROS production data were expressed as fluorescence arbitrary units. Alternatively, DHR 123 fluorescence was registered by using confocal microscopy with the same setting described above for the measurement of free cytosolic Ca^2+^ concentration.

### 2.11. Quantitative Reverse Transcriptase PCR (qRT-PCR)

Following BOLD-100 treatment, total RNA from the cells was extracted with the NucleoSpin RNAII Kit (Macherey-Nagel, Düren, Germany). cDNA then was obtained utilizing the Transcriptor First Strand cDNA Synthesis Kit (Roche Diagnostics GmbH, Penzberg, Germany).

qRT-PCR was performed in a CFX384 Real-Time PCR Detection System (Bio-Rad Laboratories, Hercules, CA, USA) utilizing the Power Sybr Green Mastermix (Bio-Rad Lab) and KiCqStart^®^ SYBR^®^ Green Primers (Table 1). Gene expression was measured by the ∆∆Ct method.

### 2.12. Western Immunoblotting

Western blot was carried out as previously described [31]. Cells were lysed in RIPA buffer (with a cocktail of phosphatase and protease inhibitors) and solubilized in Laemmli buffer.

Amounts of 20 µg protein from cell lysates, determined by BCA Protein Assay Kit (Cayman Chemicals, Ann Arbor, MI, USA), were separated by SDS–PAGE, blotted to nitrocellulose membrane, and probed with one of the following primary antibodies (dilution 1:400): anti-ERO1α (cat# 12007-1-AP, ProteinTech Group, Chicago, IL, USA), anti-GRP78/BiP (cat# ab21685, RRID: AB_2119834, Abcam, Cambridge, UK), anti-CHOP/GADD 153 (cat# ab11419, Abcam), anti-XBP1 (cat# ab37152, Abcam). Membranes were then incubated with the appropriate horseradish peroxidase-conjugated secondary antibody (Bethyl Laboratories, Montgomery, TX, USA; dilution 1:1000), developed by an ECL kit, acquired by ChemiDoc XRS (Bio-Rad Laboratories, Hercules, CA, USA), and digitized by Quantity One Imaging System (Bio-Rad). Equal loading was confirmed with anti-β-actin antibody (cat# A300-491A, Bethyl Laboratories, Montgomery, TX, USA).

### 2.13. Cell Surface Enzyme-Linked Immunosorbent Assay (ELISA)

Cell surface GRP78 (csGRP78) expression was quantified, as previously described, by using cell surface ELISA [19]. Cells were seeded in 96-well plates, treated or not with BOLD-100 for different time periods (1, 3, 24, and 48 h), rinsed with PBS, fixed with 4% formaldehyde in PBS, washed with a wash buffer (0.5 mM CaCl_2_, 1 mM MgCl_2_, and 0.1% Triton in PBS), and blocked for 30 min with 3% BSA in wash buffer. Cells were then incubated with the above anti-GRP78 antibody (diluted 1:300) in wash buffer containing 1% BSA for 2 h, incubated with horseradish peroxidase-labeled secondary antibody (Bethyl Laboratory, diluted 1:1000) for 1 h, incubated with 3,3′,5,5′tetramethylbenzidine substrate for 5 min, and read at 620 nm in the Infinite 200 Pro microplate reader (Tecan).

### 2.14. Statistical Analysis

GraphPad Prism 8 (Graphpad Software Inc., GraphPad Software, Inc., San Diego, CA, USA) was used to perform statistical analysis. Based on the data, t-test, one-way or two-way ANOVAs, followed by an appropriate correction (Dunnett’s post-test, Tukey’s test, Sidak’s multiple comparison test and the Bonferroni correction) were applied. Statistical details (value of *n*, test used, *p* value, replicates, etc) are reported in the Figure Legends.

## 3. Results

### 3.1. Cytotoxicity Assay

The cytotoxic effect of BOLD-100 on two MPM cell lines, REN and MM98, was evaluated after 24 or 48 h treatment with increasing concentrations (0–1000 µM) of BOLD-100. The MPM cell lines showed a dose-dependent effect with lower EC_50_ values, in contrast to higher EC50 values recorded for the normal immortalized mesothelial Met5A cell line, as listed in Table 2.

Considering the similarity in EC_50_ values obtained for REN and MM98 cell lines, and since the epithelioid is the most frequent histotype, we decided to perform subsequent experiments on the epithelioid REN cells using Met5A cells as a reference control cell line.

Next, to assess clinical significance we tested whether BOLD-100 impacts the long-term viability of these MPM cells. We thus conducted clonogenic assays to measure the effect of BOLD-100 on cell viability for a longer term. The cells were incubated with 100 µM BOLD-100 for 48 h, a dose chosen based on the EC_50_ value. Subsequently, the cells were washed and sparsely seeded into normal growth medium, and the growth of colonies from single cells was scored 10 days later. BOLD-100 reduced the colony formation by 50% compared to the control (Figure 1).

Upon treatment with increasing concentrations (0–500 µM) of BOLD-100 for 24 h, the induction of cytotoxicity was strongly prevented by using 10 mM caffeine (inhibitor of the inositol trisphosphate receptor, IP3R) or 50 mM NAC (N-acetyl cysteine, ROS inhibitor). In the presence of these inhibitors, the REN cell line showed 3 to 4 times higher EC_50_ values, in contrast to that obtained with BOLD-100 treatment alone (Appendix A).

### 3.2. Activation of Apoptosis

BOLD-100 has previously been shown to induce apoptotic cell death. In our model system, we reproduced these results, showing that BOLD-100 leads to the induction of apoptotic cell death, as confirmed by caspase 3/7 activation (Figure 2A). An activation of caspase 8 by ruthenium-derived anticancer compounds has also been reported [32]. The use of 100 µM BOLD-100 was able to enhance the activity of caspase 8, and this increase was abolished by the caspase-8 specific inhibitor (Figure 2B).

Undoubtedly, caspases are important in carrying out the apoptotic pathway, but several other types of non-caspase proteases, such as calpain, also play a role in the execution of this process. Calpain is a non-lysosomal cysteine protease activated by a sustained [Ca^2+^]_i_ increase [33]. To figure out calpain activation, t-BOC-loaded cells were treated with 100 µM BOLD-100 alone or in the presence of the calpain inhibitor PD150606. Treatment with BOLD-100 alone showed a doubling of calpain activation as compared to the control (CTRL), while the presence of PD150606 completely abolished the effect (Figure 2C).

It is known that the apoptotic cascade can also be activated by mitochondria and earlier studies with KP1339 (same as BOLD-100) indeed reported that BOLD-100 causes mitochondrial depolarization [34]. Thus, to address whether BOLD-100 treatment induces mitochondrial membrane destabilization in MPM cells, calcein-AM-loaded cells, in the presence of CoCl_2_, were treated in the presence or absence of 100 µM BOLD-100. The opening of the mitochondrial transition pore due to BOLD-100 exposure led to a calcein-AM release from the mitochondria into the cytosol, resulting in a reduction in fluorescence.

The significant reduction of the residual fluorescence after treatment with BOLD-100 highlighted a destabilization of mitochondrial membranes (Figure 3A). This mitochondrial destabilization was also confirmed by the variation in the mitochondrial membrane potential (ΔΨm) by staining cells with the JC-1 probe. The JC-1 dye, in healthy mitochondria, spontaneously forms red fluorescent J-aggregates, while in apoptotic cells and unhealthy mitochondria, JC-1 is unable to reach such a concentration, retaining its original green fluorescence [29]. The red/green ratio after BOLD-100 treatment was significantly lower with respect to the untreated cells. The mitochondrial destabilization was partially prevented pretreating REN cells with caffeine or NAC as described above (Figure 3B).

### 3.3. ROS Alteration after BOLD-100 Treatment

Under physiological conditions, the ER maintains an oxidizing environment due to the normal production of H_2_O_2_, as a secondary product of PDI (protein disulfide isomerase) oxidation by ERO1α (endoplasmic reticulum oxidoreductin 1 α). Under stressful conditions, the ER exhibits hyper-oxidation leading to an alteration of the ER redox state and to the diffusion of ROS in the cytoplasm, resulting in an unbalanced [GSH]:[GSSG] ratio [35].

Mesothelioma cells, compared to their non-tumor counterpart, have an altered level of ROS, and in particular a high level of superoxide anion due to a high level of NOX4 expression [36], leading mesothelioma cells to exhibit a higher level of ER stress than normal mesothelial cells, as previously highlighted [19]. The treatment of MPM cells with BOLD-100 immediately enhances the intracellular ROS level, in contrast to that observed in mesothelial cells (Figure 4A). Additionally, after 4 h treatment with increasing doses of BOLD-100, no variation of ROS in Met5A cells was observed, whereas REN cells exhibit a dose-dependent increase in cytosolic ROS. Indeed, the ROS induction achieved with the highest concentration of BOLD-100 is comparable to that achieved with TBHP (tert-Butyl hydroperoxide) (Figure 4B). This increase of ROS was partially prevented by pretreating the cells with caffeine (Figure 4C).

The expression of ERO1α was then evaluated by western blot analysis. Comparing ERO1α basal level in Met5A and REN cells, we observed a basal expression significantly higher in REN cells by about 4-fold. By treating REN cells with 100 µM BOLD-100, ERO1α expression level increased around 2-fold from baseline (Figure 4D).

### 3.4. Intracellular Ca^2+^ Concentration ([Ca^2+^]_i_) Variations

Cell survival and death are mainly regulated by Ca^2+^ that activates or inactivates different regulatory proteins such as molecular chaperones, enzymes, or transcriptional factors. Several authors showed how a calcium homeostasis disorder can lead to various types of cell death in cancer cells [37,38] and how this modified Ca^2+^ homeostasis weaves together with the ER redox state alteration [39]. For this reason, a confocal time-lapse analysis was performed to evaluate the [Ca^2+^]_i_ variation induced by BOLD-100. Based on the cytotoxicity results, we chose 100 µM BOLD-100 as the working concentration for this assay. Consistent with the viability test, we observed a variation in the [Ca^2+^]_i_ only in the REN cells, while no effect in the Met5A cells was observed. Such a variation is delineated by a peak phase immediately after the addiction of BOLD-100, followed by a plateau phase (Figure 5A). The confocal analysis was then carried out in continuum in the same treatment conditions, and any significant oscillation in the [Ca^2+^]_i_ was registered (Figure 5B).

The endoplasmic reticulum (ER) plays a master role in many fundamental functions for cell survival, and several of these are tightly related with the ER Ca^2+^ level. An alteration of the ER homeostasis, as with a shift in the redox state, could lead to an ER stress condition [40]. As a matter of fact, pretreatment with the ROS scavenger NAC resulted in complete abrogation of the calcium peak observed after BOLD-100 addition (Figure 6A). In order to determine if ER is the source of calcium observed into the cytosol during [Ca^2+^]_i_ variation due to BOLD-100 treatment, the time-lapse observation for BOLD-100 treatment was thus repeated after 30 min of pretreatment with 5 µM thapsigargin, a non-competitive inhibitor of the sarco/endoplasmic reticulum Ca^2+^ ATPase (SERCA) [31]. In the presence of thapsigargin, used as a pretreatment to induce ER calcium depletion, the [Ca^2+^]_i_ peak was completely reduced, confirming that the ER was the Ca^2+^ source (Figure 6B). The most abundant and ubiquitously expressed ER Ca^2+^-release channel is the IP3R (1,4,5-triphosphate receptor). Its activation leads to an IP3-induced Ca^2+^ release (IICR) that plays a central role in the regulation of many cellular processes, including cell survival or cell death. Moreover, the presence of a hyper-oxidizing environment indicates a sensitization of the IP3R and consequently an increase of the IICR [40,41]. Thus, PLC-IP3 pathway inhibitors in the presence of BOLD-100 were utilized to repeat the Ca^2+^ observations. REN cells were pre-treated with 50 µM 2-APB or 10 mM caffeine (inhibitors of the IICR) or with 10 µM U73122 (a potent phospholipase C (PLC) inhibitor) [25]. In the presence of all these inhibitors, the [Ca^2+^]_i_ peak, induced by BOLD-100 treatment, undergoes an almost complete abrogation (Figure 6C). Another resident protein in the ER membrane is the translocon (TLC), a protein complex that in physiological conditions works as a protein-conducting channel; it is not clear if it plays a role in the basal ER Ca^2+^ regulation permitting Ca^2+^ leakage. Moreover, under ER stress conditions, a massive Ca^2+^ leakage was observed through the translocon [42]. To prove its involvement in the generation of the [Ca^2+^]_i_ peak after the BOLD-100 addition, the confocal time-lapse observation was repeated in the presence of 200 µM anisomycin (30 min pretreatment), that inhibits the ER Ca^2+^ leak through TLC [43]. Keeping TLC in a closed conformation with anisomycin, only little [Ca^2+^]_i_ variations were recorded (Figure 6D). An increase in [Ca^2+^]_i_ in the cytosol was observed upon treating REN cells with puromycin (2 µM) an activator of TLC [42], which was comparable to the increase observed by BOLD-100 (Appendix A). All the inhibitors utilized significantly block the BOLD-100 calcium rise, but comparing the inhibitory effects, PLC-IP3R inhibitors showed a higher inhibitory effect than the TLC inhibitor, suggesting the predominant role of the PLC-IP3R pathway (Figure 6E).

### 3.5. GRP78 Variation and ER Stress Response

Altered ER homeostasis due to a shifted redox state leads to a pro-apoptotic condition. The overexpression of ERO1α determines an increase in the ROS production, in particular H_2_O_2_, and is able to activate an ER stress response [39]. After treatment with BOLD-100, we observed a 50% increase in the expression of ERO1α in REN cells. Thus, we measured the activation of the UPR/ER stress response, focusing our attention on the GRP78 downstream effectors closely involved in the ER-stress-induced cell death: the ATF4/CHOP, and XBP1 axes [19].

The expression of GRP78, CHOP, and XBP1 were evaluated at both mRNA and protein levels. The mRNA levels were assessed after BOLD-100 treatment for 1 and 24 h. CHOP and XBP1 showed a time dependent increase, whereas GRP78 showed a slight decrease after 24 h treatment (Figure 7A). For the evaluation of protein levels, we treated for 24 and 48 h, and at both time points GRP78 showed a time dependent decrease, consistent with the already-known effect of BOLD-100, whereas the down-stream effectors, CHOP and XBP1, displayed an increased expression (Figure 7B).

In physiological conditions, GRP78 works as an ER chaperone, but during an ER stress disorder, such as ER Ca^2+^-depletion, it could escape from ER retention leading to a translocation to the plasma membrane conferring to cancer cells an increased malignant behavior [44,45] and activating pro-survival or pro-apoptotic pathways. Evaluating by cell surface ELISA the csGRP78 expression, after BOLD-100 treatment in a time period ranging from 1 to 48 h, we observed an increase at 1 h, reaching a peak after 3 h, and then a decrease (Figure 7C).

## 4. Discussion

Malignant pleural mesothelioma is considered to be a rare cancer worldwide, but it is characterized by high incidence in some regions that appear as hotspots [46]. The disease is identified mostly in the advanced stages, at which point there are limited efficient therapeutic treatments beyond standard chemotherapeutics, underscoring the need for novel therapies [46].

Currently, the standard therapy for MPM is based on the use of a combination strategy (platinum–pemetrexed chemotherapy [47]) where two different drugs are combined. Thus, the idea of combining a novel agent with a drug already in use could be a good strategy.

Accordingly, based on our previous data indicating a high expression of GRP78 in mesothelioma REN and MM98 cells [19], we investigated the mechanism of action of the clinical-stage drug BOLD-100 on MPM cells, to disclose the potential of an innovative drug that could be combined with the standard-of-care drugs in the clinic for the treatment of MPM.

BOLD-100 is a ruthenium-based small molecule therapeutic, currently being tested in a phase 1b/2 clinical trial in combination with the chemotherapy FOLFOX (fluorouracil, oxaliplatin and leucovorin) for the treatment of advanced gastrointestinal cancers (NCT04421820). Moreover, BOLD-100 has also demonstrated a strong potential to synergize with checkpoint inhibitors in different preclinical models. Zhang et al. showed the strong combinational potential of BOLD-100 within organosilica nanoparticles with immuno-oncology agents, demonstrating also that BOLD-100 can induce immunogenic cell death [48].

BOLD-100 has a complex mechanism but appears able to selectively prevent the upregulation of GRP78 triggered by stress in cancer cells and to activate the unfolded protein response (UPR), with minimal impact on normal cells [12,49].

First, we evaluated the cytotoxicity on MPM or mesothelial cells highlighting a more toxic behavior on MPM cells than on non-cancerous mesothelial Met5A cells. EC_50_ values obtained are in a similar range to those shown in different tumor cell lines [49]. Furthermore, BOLD-100 treatment also prevented the ability of MPM cells to form new colonies, indicating a long-term cytotoxic effect of the drug [27]. The BOLD-100 cytotoxicity was almost completely prevented by caffeine and NAC, suggesting the involvement of ER calcium and ROS in the induction of cell death.

Cell death observed after BOLD-100 treatment occurs in an apoptotic way by the activation of caspase 3/7 as well as caspase 8. These data confirmed the ability of ruthenium drugs, as reported by other authors, to induce also in MPM cells the apoptotic cascade [21]. In our cellular model, we were able to show that the apoptotic pathway, boosted by BOLD-100 treatment, is carried out not only by caspases, but also by the activation of the non-lysosomal cysteine proteases calpains [33] and by the destabilization of the mitochondrial membrane. The latter is highlighted by the opening of the mitochondrial transition pore and the variation of the mitochondrial membrane potential. These data, along with those of cytotoxicity obtained using caffeine and NAC, suggest again that the involvement of ROS and ER stress mediated calcium release in the BOLD-100 mechanism of action in MPM cells.

Mesothelioma cells are characterized by an altered level of ROS and a high level of ER stress [19,36]; BOLD-100 treatment magnifies this constitutive condition. Ruthenium-based drugs are able to induce an increase in ROS levels in colon carcinoma cells [50]; in MPM cells, we observed a slight increase in intracellular ROS levels immediately after BOLD-100 addition and then, after 4 h a dose-dependent increase was observed. In the ER lumen, the resident protein ERO1α is a source of ROS, producing H_2_O_2_ as a secondary product of PDI oxidation [40]. MPM cells display a high level of ERO1α as compared to normal mesothelial cells, highlighting a source of ROS that contributes to the rise of the ER stress level in basal conditions. In this context, the treatment of mesothelioma REN cells with BOLD-100 induces the doubling of ERO1α protein level. Therefore, with a further increase in ROS upon the administration of BOLD-100, the physiological balance of the ER redox environment is highly compromised, leading to the diffusion of ROS into the cytosol [35]. The intracellular rise in ROS could be due to ER, but also due to mitochondrial release after the destabilization of mitochondrial membrane potential, as the ER and mitochondrial crosstalk plays a crucial role in the redox state and response [39]. The contact point between mitochondria and the ER, called MAM (mitochondria-associated ER membrane), contributes to the movement of Ca^2+^ from the ER to the mitochondria [51,52], and this flux is mediated by the IP3R on the ER membrane, VDAC (voltage-dependent anion channel) on the outer mitochondrial membrane, and the MCU (mitochondrial calcium uniporter) on the inner mitochondrial membrane [53,54]. When the increase of Ca^2+^ into the mitochondrial matrix reaches a critical threshold, the MPTP (mitochondrial permeability transition pore) opens, triggering the apoptotic cascade [55]. The inhibition of the calcium flux from the ER to the mitochondria through IP3R abrogated the detectable mitochondrial membrane potential destabilization and cell death observed after BOLD-100 treatment.

Considering the key role of the ER–mitochondria crosstalk in the induction of apoptosis, we evaluated the variation in the [Ca^2+^]_i_ after the addition of BOLD-100. We observed a Ca^2+^ peak in MPM REN cells, but not in normal mesothelial Met5A cells or in REN cells pretreated with thapsigargin, confirming that this rise in the cytoplasmatic concentration is due to an outflow from the ER. IP3R- and IP3-induced Ca^2+^ release (IICR) are main players in the regulation of cell fate. Therefore some PLC-IP3 pathway inhibitors were utilized in Ca^2+^ observations and with all these inhibitors, the [Ca^2+^]_i_ peak, induced by BOLD-100 treatment, undergoes an almost complete abrogation. The alteration of the redox ER environment due to the overexpression of ERO1α sensitized IP3R and consequently caused an increase of the IICR [27,28]. The redox-sensitive binding of ERp44, an ER chaperone belonging to the thioredoxin family, controls IP3R inhibiting the IICR and protecting cells against apoptosis. The hyper-oxidizing ER environment may disrupt the ERp44-IP3R binding, and in particular the oxidized channel is aberrantly activated, resulting in ER calcium depletion [56,57]. Moreover, Hammadi et al. [42] previously demonstrated the Ca^2+^ leakage from the ER through the translocon (TLC), a protein complex present in the ER membrane and involved in protein translocation, acting as a calcium leak channel during the ER stress condition. When the TLC was maintained in a closed conformation by using anisomycin, despite the presence of BOLD-100, the increase in cytosolic [Ca^2+^] was not observed, whereas a cytosolic Ca^2+^ increase of similar intensity as that achieved with BOLD-100 treatment was obtained with puromycin (an activator of TLC).

These data suggest that BOLD-100 causes an abnormal release of calcium into the cytoplasm mediated by a leak through IP3R and TLC. The latter, considering the ability of some Ru-based compounds to inhibit the activity of SERCA [58], determines the maintenance of a higher [Ca^2+^]_i_ with respect to physiological conditions, contributing in a significant way to the cell death.

A characteristic trait of MPM cells is the constitutive deregulation of the ER homeostasis and high UPR signaling, with respect to mesothelial cells, which confer resistance to standard therapy. However, this feature could be utilized as a therapeutic target by further increasing the ER stress [19,59]. The intrinsic MPM ER stress condition is heightened by the instability of the redox state determined by BOLD-100 treatment. Therefore, the increase of H_2_O_2_, produced as a by-product of ERO1α activity results in an accumulation of misfolded proteins and can stimulate a UPR cascade causing at least apoptosis [60].

We already described a high GRP78 expression level in MPM cells with respect to the mesothelial cell [19,24], but after BOLD-100 treatment we observed a time-dependent decrease, despite a small increase in the mRNA level immediately after treatment. The major downstream pro-apoptotic factor is CHOP, which in physiological condition shows a low expression level that is enhanced under ER stress. The PERK/ATF4/CHOP axis plays a key role in the induction of cell death, but the mature XBP1 protein is also involved in the regulation of CHOP expression [61].

Moreover, CHOP can also contribute to the oxidative stress by inducing the activation of ERO1α that in turn causes an IP3R-mediated Ca^2+^ leak, leading to the activation of pro-apoptotic pathways [60]. The treatment of REN cells with BOLD-100 led to an increase of XBP1 and CHOP both at the mRNA and at the protein level.

The ER chaperone and stress sensor, GRP78 is known to be found on the cell surface or secreted in the extracellular environment [62]. csGRP78 functions like a signal receptor, transmitting some extracellular signals into cells. In the secreted soluble form, GRP78 was recently identified as a pro-apoptotic ligand of csGRP78, able to trigger caspase-mediated apoptosis in stressed pancreatic beta cells. Therefore, csGRP78 acts as a death receptor for the secreted form, which as a self-ligand activates the apoptotic pathway leading to cell death [62]. We previously observed in REN cells the presence of csGRP78 and also the secreted form in the conditioned medium [19,24]. After BOLD-100 treatment, we observed an increase in GRP78, but part of this translocates to the plasma membrane within 1 to 3 h of treatment. The presence of csGRP78, considering the secreted form as ligand, could cooperate to start the apoptotic pathway that we observed after BOLD-100 treatment.

## 5. Conclusions

In conclusion, this is the first demonstration of the mechanism of action of BOLD-100 involved in mesothelioma cell death induction, which can be summarized as seen in Figure 8:

Treatment with BOLD-100 determines a disruption of ER homeostasis:
oBOLD-100 causes an increase of ROS as a consequence of the induction of ERO1α overexpression;oThe rise in ROS and ERO1α levels destabilize ER calcium regulation, leading to a massive release into the cytosol either through TLC and IP3R leakage, but also a Ca^2+^ overload into the mitochondria by way of MAM.
Following ER homeostasis loss, UPR activation starts:
oGRP78 firstly increases and then decreases in a time-dependent manner;oGRP78 translocates to the plasma membrane in a time-dependent manner immediately after the treatment;oXBP1 and CHOP increase in a time-dependent manner.
All these pathways converge in the induction of apoptosis in MPM cells through different ways:
oThe activation of caspase3/7 and caspase 8;oThe destabilization of mitochondrial membrane potential;oThe activation of calpain.


Taken together, our observations suggest that further in vivo studies are urgently needed to explore the application of BOLD-100 as single agent, or in combination with other drugs [49], to fight malignant pleural mesothelioma.

## Figures and Tables

**Figure 1 cancers-14-04126-f001:**
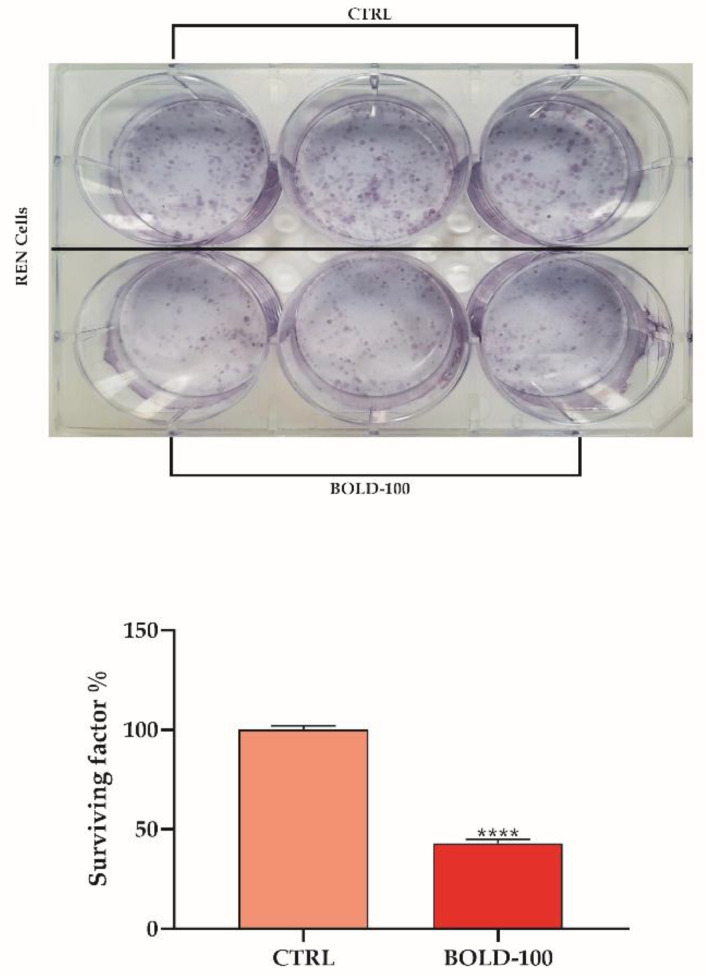
BOLD-100 reduced colony formation in REN cells. Colony formation ability measured in cells exposed for 48 h to BOLD-100. Data are means ± SD from five independent treatments, and indicated as surviving factor %. Asterisks on bars indicate statistical differences (**** *p* < 0.0001, unpaired *t* test).

**Figure 2 cancers-14-04126-f002:**
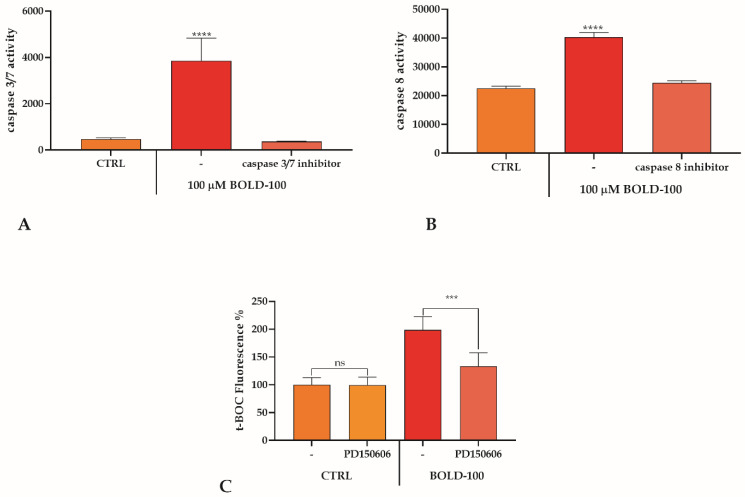
BOLD-100-induced apoptosis in REN cells. (**A**). Caspase 3/7 activation measured in cells exposed for 1 h to BOLD-100 in the presence or absence of a caspase 3/7 inhibitor. Data are means ± SD from five independent treatments and are indicated as caspase 3/7 activity. Asterisks on bars indicate statistical differences (**** *p* < 0.0001, one-way ANOVA followed by Tukey’s test). (**B**). Caspase 8 activation measured in cells exposed for 1 h to BOLD-100 in the presence or absence of a caspase 8 inhibitor. Data are means ± SD from five independent treatments and are indicated as caspase 8 activity. Asterisks on bars indicate statistical differences (**** *p* < 0.0001, one-way ANOVA followed by Bonferroni correction). (**C**). Calpain activity in cells exposed for 1 h to BOLD-100 in the presence or absence of PD150606. Data are means ± SD from five independent treatments and are expressed as fluorescence%. Statistics indicated significant differences between treatment in the presence or absence of inhibitors (*** *p* < 0.001, ns not significant, one-way ANOVA followed by Bonferroni correction).

**Figure 3 cancers-14-04126-f003:**
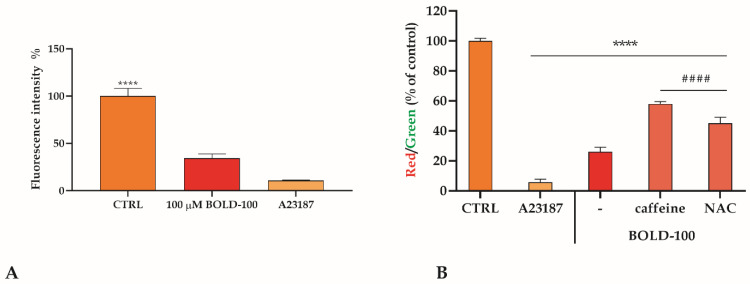
BOLD-100 determines mitochondrial destabilization in REN cells. (**A**). Mitochondrial membrane destabilization evaluated in cells after 1 h exposure to BOLD-100 or ionophore A23187 used as a positive control. Data are means ± SD from five independent treatments, and are expressed as fluorescence intensity%. Statistics indicate differences with respect to the CTRL (**** *p* < 0.0001, one-way ANOVA followed by Bonferroni correction). (**B**). Mitochondrial membrane potential (ΔΨm) variation assessed in cells after 1 h exposure to BOLD-100 or ionophore A23187 used as a positive control. Data are means ± SD from five independent treatments, and are expressed as a red/green ratio with respect to the % of control. Statistics indicated significant differences with respect to the CTRL (**** *p* < 0.0001, one-way ANOVA followed by Dunnett’s post-test) or differences between treatment (#### *p* < 0.0001, one-way ANOVA followed by Sidak’s multiple comparison test).

**Figure 4 cancers-14-04126-f004:**
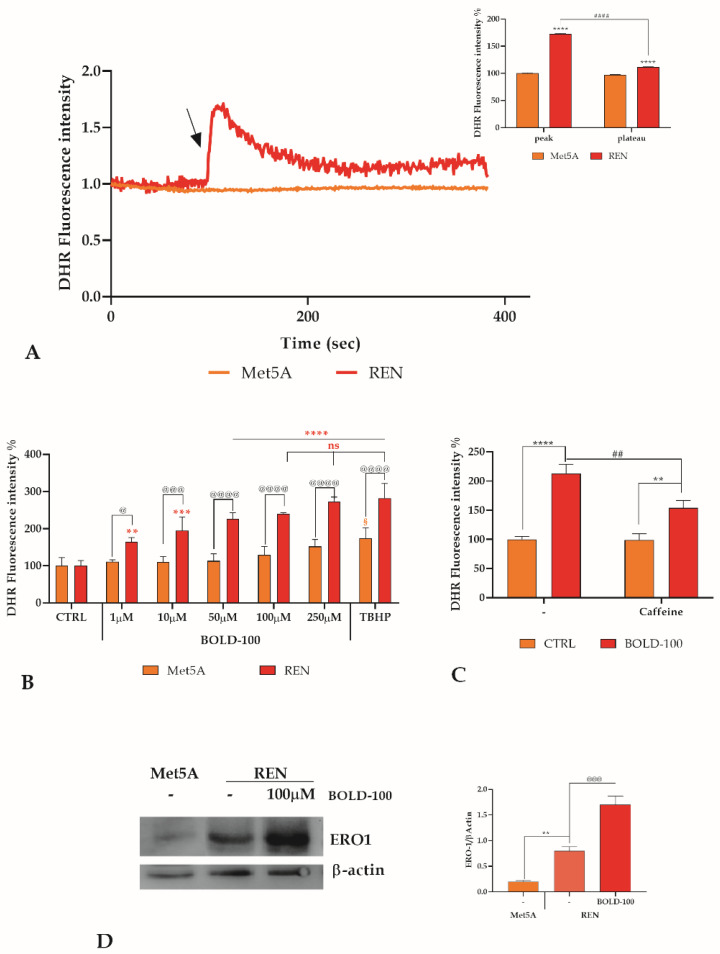
ROS alteration after BOLD-100 treatment. (**A**). ROS production evaluated as DHR fluorescence intensity after BOLD-100 treatment recorded at 1 sec intervals in Met5A and REN cell lines. The arrow shows the BOLD-100 addition after 80 sec. Data are means of ROS traces recorded in different cells. Number of cells: 40 cells from three experiments for each cell line. Insert. Mean ± SEM of ROS peak response. Number of cells is as before. Statistics indicate differences between Met5A and REN cell line at the peak or at the plateau level (**** *p* < 0.0001, two-way ANOVA followed by Tukey’s test), or differences between peak and plateau in REN cell line (#### *p* < 0.001, two-way ANOVA followed by Tukey’s test). (**B**). ROS production evaluated as DHR fluorescence intensity recorded at 4 h in control cells (CTRL), or in cells incubated with different concentrations of BOLD-100 or with TBHP used as positive control. Data are means ± SD of DHR-123 fluorescence measured in arbitrary units; *n* = 16 microplate wells from two different experiments. Statistics indicate differences between Met5A and REN cell line (@ *p* < 0.05, @@@ *p* < 0.001, @@@@ *p* < 0.0001). § indicates differences between CTRL and TBHP in Met5A cell line (§ *p* <0.01). Asterisks indicate differences between CTRL and BOLD-100 or TBHP treatment in REN cell line (** *p* < 0.01, *** *p* < 0.001, **** *p* < 0.0001, one-way ANOVA followed by Dunnett’s post-test). Differences between highest BOLD-100 concentrations and positive control are not significant (ns not significant, one-way ANOVA followed by Sidak’s multiple comparison test). (**C**) ROS production evaluated as in B. Statistics indicate differences between CTRL and BOLD-100 treatment (** *p* < 0.01, **** *p* < 0.0001, one-way ANOVA followed by Dunnett’s post-test) or between BOLD-100 treatment in presence or not of caffeine (## *p* <0.01 one-way ANOVA followed by Sidak’s multiple comparison test). (**D**) ERO1α protein expression in mesothelial Met5A cells compared to REN cells, and variation in REN cells after BOLD-100 treatment. Blots shown are representative of three; lanes were loaded with 20 μg of proteins, probed with anti-ERO1α rabbit poly-clonal antibody and managed as described in Materials and Methods. The same blots were stripped and re-probed with anti-beta-actin (β-ACT) poly-clonal antibody. The ratio between the ERO1α band and the corresponding β-actin band used as loading control was utilized for the quantification of band intensities. Data are means ±SD derived from three independent samples. Statistics indicate differences between Met5A and REN cell line (** *p* < 0.01, one-way ANOVA followed by Bonferroni correction) or differences in REN cell line after treatment or not with 100 μM BOLD-100 (@@@ *p* < 0.001, one-way ANOVA followed by Bonferroni correction). The uncropped blots are shown in Appendix A.

**Figure 5 cancers-14-04126-f005:**
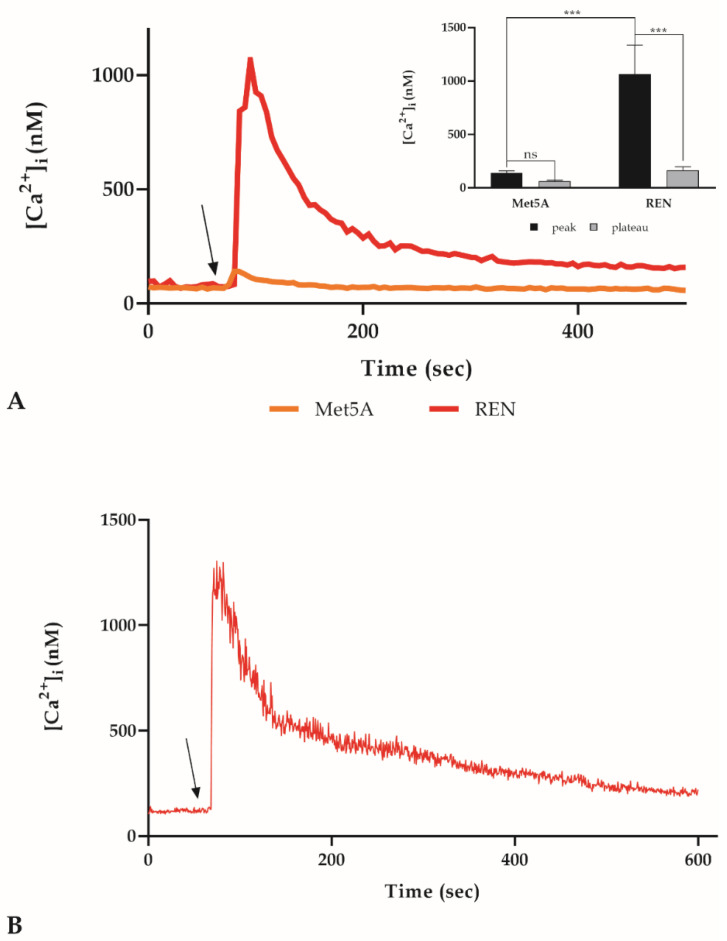
Variation of [Ca^2+^]_i_ induced by BOLD-100. (**A**). [Ca^2+^]_i_ modifications after BOLD-treatment recorded at 5 sec intervals in Met5A and REN cell lines. The arrow shows the BOLD-100 addition after 60 sec. Data are means of [Ca^2+^]_i_ traces recorded in different cells. Number of cells: 40 cells from three experiments for each cell line. Insert. Means ± SEM of Ca^2+^ peak response. Number of cells as before. Statistics indicate differences between Met5A and REN cell line at the peak or at the plateau level (*** *p* < 0.001, two-way ANOVA followed by Bonferroni correction), or differences between peak and plateau for each cell line (*** *p* < 0.001; ns not significant, two-way ANOVA followed by Bonferroni correction). (**B**). [Ca^2+^]_i_ variations induced by BOLD-100 recorded at 1 sec intervals. Data are means ± SEM of [Ca^2+^]_i_ traces recorded in 40 different cells.

**Figure 6 cancers-14-04126-f006:**
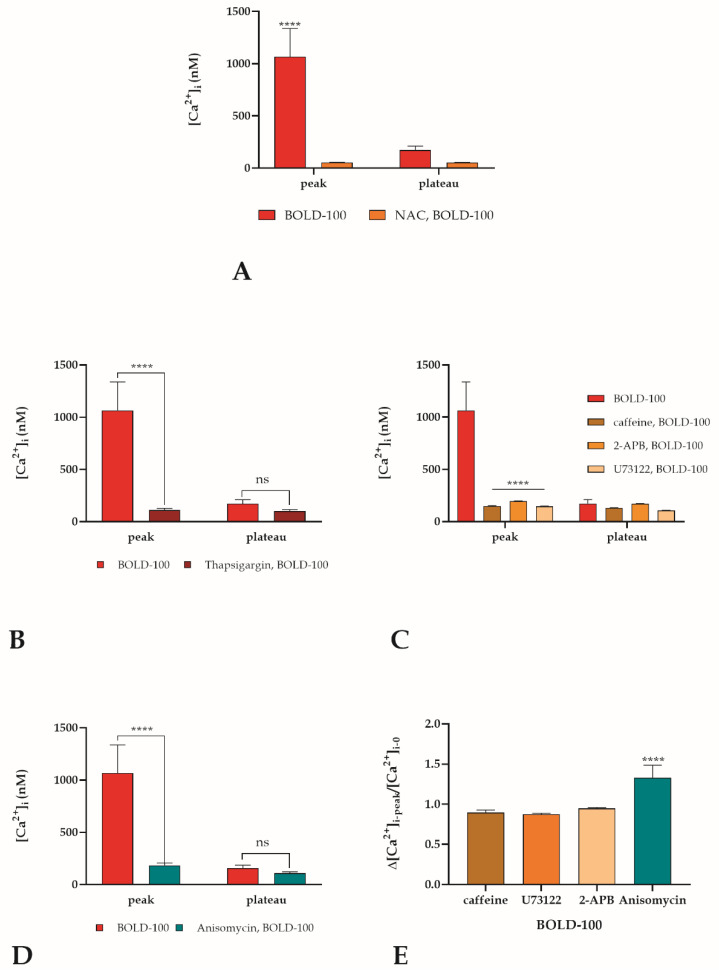
Origin of cytosolic Ca^2+^ after treatment with BOLD-100. [Ca^2+^]_i_ variation after BOLD-treatment recorded at 5 sec intervals in REN cells in the presence or absence of: (**A**) 50 mM NAC; (**B**) 5 µM thapsigargin; (**C**) 50 µM 2-APB or 10 mM caffeine or 10 µM U73122; (**D**) 200 µM anisomycin. Data are means ± SEM of Ca^2+^ peak response recorded in different cells. Number of cells: 40 cells from three experiments for each condition. Statistics indicate differences at peak or plateau level for each condition (**** *p* < 0.0001; ns not significant, two-way ANOVA followed by Bonferroni correction). (**E**) Comparison between inhibitors in (**C**,**D**) expressed as Δ[Ca^2+^]_i-peak_/[Ca^2+^]_i-0_; Data are means ± SEM of Ca^2+^ Δ[Ca^2+^]_i-peak_/[Ca^2+^]_i-0_ recorded in different cells. Number of cells: 30 cells from three experiments for each condition. Statistics indicate differences between different conditions (**** *p* < 0.0001, one-way ANOVA followed by Dunnett’s post-test).

**Figure 7 cancers-14-04126-f007:**
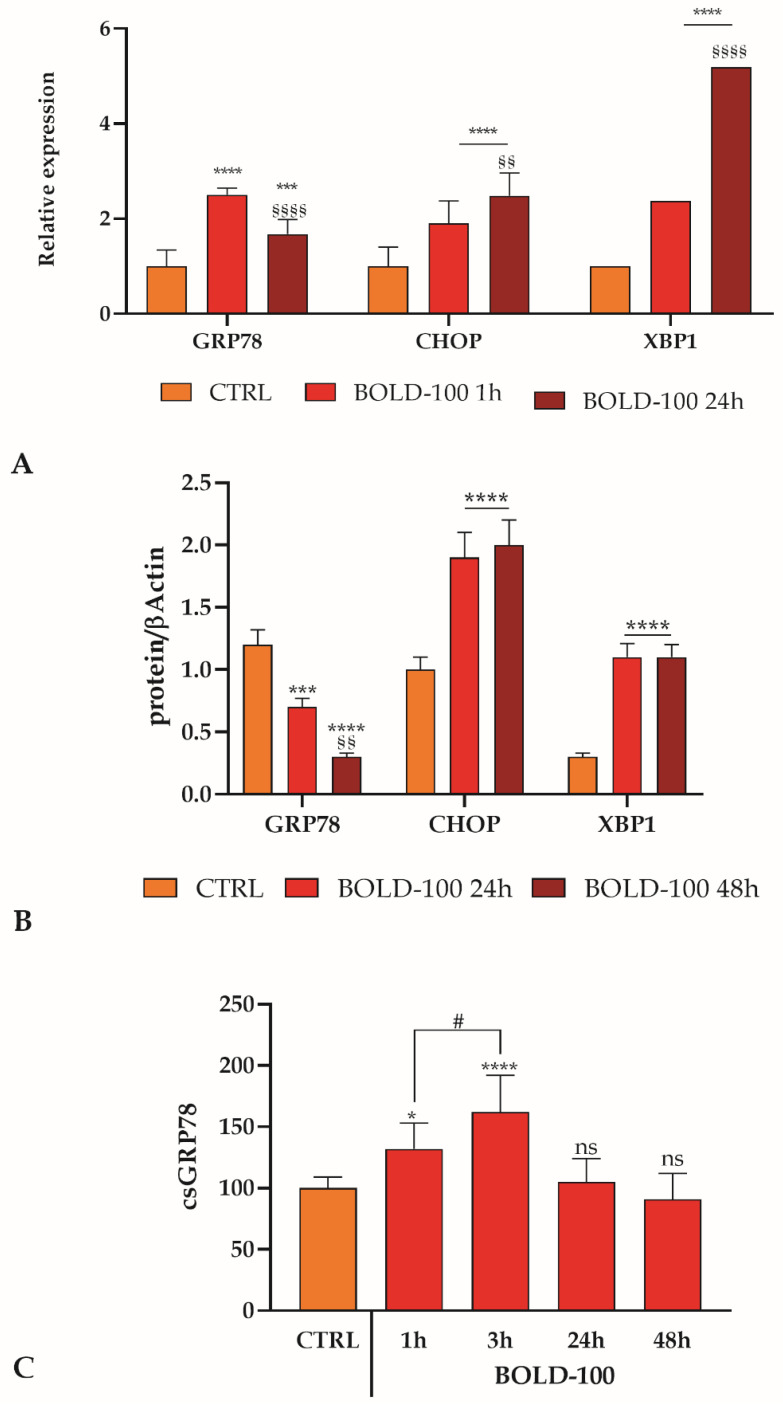
UPR activation in BOLD-100 treated REN cells. (**A**). mRNA quantity of GRP78, CHOP and XBP1 transcripts determined by qRT-PCR after BOLD-100 treatment. Data are mean relative expression ± SD (*n* = 3). Statistics indicate, for each mRNA group, differences with respect to the CTRL (*** *p* < 0.001, **** *p* < 0.0001, two-way ANOVA followed by Tukey’s test) or between different times of BOLD-100 treatment (§§ *p* < 0.001, §§§§ *p* < 0.0001, one-way ANOVA followed by Tukey’s test). (**B**). GRP78, CHOP, and XBP1 protein expression variation in REN cells after BOLD-100 treatment. Blots shown are representative of three; lanes were loaded with 20 μg of proteins, probed with anti-GRP78 or anti-CHOP or anti-XBP1 rabbit poly-clonal antibody and managed as described in Materials and Methods. The same blots were stripped and re-probed with anti-beta-actin (β-ACT) rabbit poly-clonal antibody. The ratio between the protein of interest band and the corresponding β-actin band used as loading control was utilized for the quantification of band intensities. Data are means ± SD derived from three independent samples. Statistics indicate, for each protein level group, differences with respect to the CTRL (*** *p* < 0.001, **** *p* < 0.0001, two-way ANOVA followed by Tukey’s test) or between different times of BOLD-100 treatment (§§ *p* < 0.01, one-way ANOVA followed by Tukey’s test). (**C**). Quantification of csGRP78 expression in REN cell by cell surface ELISA after BOLD-100 treatment. Data are mean expression % ± SD (*n* = 3). Statistics indicate differences with respect to the CTRL (* *p* < 0.05; **** *p* < 0.0001, ns not significant, one-way ANOVA followed by Dunnett’s post-test) or between 1 and 3 h time points of BOLD-100 treatment (# *p* < 0.05, one-way ANOVA followed by Sidak’s multiple comparison test).

**Figure 8 cancers-14-04126-f008:**
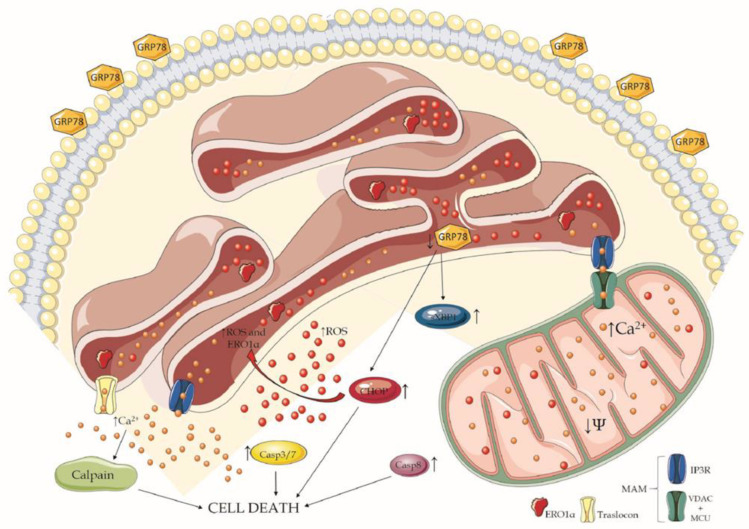
Diagram representing the mechanism of BOLD-100 toxicity on mesothelioma cells. The schematic art pieces utilized were obtained by Servier Medical art (http://smart.servier.com/, accessed on 18 May 2022). Servier Medical Art by Servier is licensed under a Creative Commons Attribution 3.0 Unported License.

**Table 1 cancers-14-04126-t001:** List of primers used in qRT-PCR analysis.

Gene	Sequences
GRP78	TGTTCAACCAATTATCAGCAAACTC TTCTGCTGTATCCTCTTCACCAGT
CHOP	AGAACCAGGAAACGGAAACAGA TCTCCTTCATGCGCTGCTTT
sXBP1	CTGAGTCCGAATCAGGTGCAG ATCCATGGGGAGATGTTCTGG
usXBP1	CAGCACTCAGACTACGTGCA ATCCATGGGGAGATGTTCTGG
β-actin	TCCCTGGAGAAGAGCTACGA AGCACTGTGTTGGCGTACAG

**Table 2 cancers-14-04126-t002:** BOLD-100 effect on cell viability evaluated in terms of EC_50_ at 24 or 48 h. Each value comes from the results of three independent experiments. In brackets, the 95% confidence interval. Concentrations are indicated as µM.

Cell Line	EC_50_ at 24 h	EC_50_ at 48 h
REN	111 (81–152)	71 (50–102)
MM98	219 (176–273)	90 (70–116)
Met5A	572 (460–712)	399 (249–636)

## Data Availability

The data presented in this study are available on request from the corresponding author.

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
