# Peer review of "ER Stress Response and Induction of Apoptosis in Malignant Pleural Mesothelioma: The Achilles Heel Targeted by the Anticancer Ruthenium Drug BOLD-100"

_cancers, 2022, doi:10.3390/cancers14174126_

Round 1

Reviewer 1 Report

The manuscript submitted by Ranzato et al, have shown that BOLD-100 shows cytotoxicity in MPM cells and increases the oxidative stress and deregulates the calcium homeostasis leading to cell stress and, ultimately, to cell death. This study is based on their previous finding of importance of GRP78 in MPM survival. While the subject of this investigation is important, the following points may help to significantly improve the quality of the manuscript and the value of the data within the MS

1.     The authors claims that there is no efficient therapeutic treatment, this statement mentioned in the abstract is not completely true as Zucali et al and others have recently reviewed new drugs and combination of drugs for malignant pleural mesothelioma (doi: 10.21037/jtd.2017.10.131)

2.       Though, authors have mentioned in the introduction that BOLD-100 is in a clinical-stage therapeutic that has previously been tested in a phase 1 clinical trial as a monotherapy in advanced cancers, however, if this is the only reason of choosing BOLD-100 for this study, then it does not seem to be the very plausible reason and more explanation is needed

3.       In the results section the cytotoxicity and colony formation assays, no images for the results have been presented and direct quantification has been shown which does not seem to be convincing without the actual images.

4.       Table2 presents the EC50 values for each cell in which REN and Met5A have shown 70 & 399 µM EC50 respectively for 48 hours, however in colony formation results it reduces the colony formation 50 % which further creates doubts in absence of colony formation images for these results.

5.       Authors have shown activation of apoptosis via activation of caspases using fluorescence-based assay, though this is a plausible experiment to perform, however, fluorescent based assay have some limitation so one would want to confirm this result with another set of experiment like WB for different caspases3/7 and 8 in presence and absence of BOLD-100 and inhibitors.

6.       Figure 4D, represents the WB images in which endogenous control actin seems to be very low for 100 µg of protein as mentioned in the methods section.

7.       Reference missing for calpain activity measurement assay

Minor comments

Some grammar and typo error throughout the MS some of these mentioned below

3,7% paraformaldehyde (PFA) should be 3.7 % paraformaldehyde (PFA)

Is it 5% crystal violet or 0.5 % ?

Author Response

1. The authors claims that there is no efficient therapeutic treatment, this statement mentioned in the abstract is not completely true as Zucali et al and others have recently reviewed new drugs and combination of drugs for malignant pleural mesothelioma (doi: 10.21037/jtd.2017.10.131)

We thank the reviewer for the opportunity to cite this review and to better define therapeutical options for mesothelioma.

  1. Though, authors have mentioned in the introduction that BOLD-100 is in a clinical-stage therapeutic that has previously been tested in a phase 1 clinical trial as a monotherapy in advanced cancers, however, if this is the only reason of choosing BOLD-100 for this study, then it does not seem to be the very plausible reason and more explanation is needed

Due to GRP78 involvement in mesothelioma biology, as we previously explored, we decided to test if a drug, i.e. BOLD-100 that impact GRP78 behavior, can interfere with mesothelioma growth.

In particular, our idea is not to propose BOLD-100 as monotherapy, but in combination with classic drugs and approaches utilized for mesothelioma. In fact, BOLD-100 is showing promising results in combination with FOLFOX (fluorouracil, leucovorin and oxaliplatin) for gastrointestinal cancers (NCT04421820) and in synergy with checkpoint inhibitors.

  1. In the results section the cytotoxicity and colony formation assays, no images for the results have been presented and direct quantification has been shown which does not seem to be convincing without the actual images

We have inserted images of colony formation assay

  1. Table2 presents the EC50 values for each cell in which REN and Met5A have shown 70 & 399 µM EC50 respectively for 48 hours, however in colony formation results it reduces the colony formation 50 % which further creates doubts in absence of colony formation images for these results.

We have inserted images of colony formation assay to corroborate colony formation results. Anyway in the colony formation graph Met5A cells are not considered, the difference highlighted is between REN cells without treatment and REN cells after 48 h treatment with BOLD-100.

  1. Authors have shown activation of apoptosis via activation of caspases using fluorescence-based assay, though this is a plausible experiment to perform, however, fluorescent based assay have some limitation so one would want to confirm this result with another set of experiment like WB for different caspases3/7 and 8 in presence and absence of BOLD-100 and inhibitors.

Determination of caspase activation can be performed in different ways. The detection of activated caspases could be performed by immunoblotting. This is a method by which the processing of a particular pro-caspase is studied by detecting (with a specific antibody) the appearance of small or large subunit of the mature caspase. The main limitation of the immunoblot technique is the requirement for high quality antibodies reacting specifically with individual caspases, as well as immunoblotting is mainly a qualitative assay (Kohler et al., Journal of Immunological Methods 2002).

One of the commonly used ways to detect caspase activity is to measure the cleavage of synthetic substrates upon their incubation with lysates of apoptotic cells. The substrate is cleaved by caspases recognizing the substrate cleavage site, thereby giving rise to increased color or fluorescence intensity.

The intensity of the signal is proportional to the amount of cleaved substrate, which in turn, is dependent on caspase activity and is related to the percentage of apoptotic cells in a cell population.

Detection of caspase activity by cleavage of synthetic substrates is a commonly used method and has several advantages. It is a quantitative, convenient, relatively fast and sensitive method.

So, we decided to study the apoptotic pathway involvement in BOLD-100 toxicity in our cell model using fluorescent synthetic substrates.

In order to avoid redundancy in the sequence specificity of caspases for substrates, i.e. many caspases have the ability to cleave the same substrates, at least in vitro, although with different efficiency, it is possible to use, as we done, specific caspase inhibitors. In particular, we used a caspase 3/7 inhibitor (N-Ac-DEVD-CHO) and a specific inhibitor of caspase-8 (Z-IETD-FMK).

  1. Figure 4D, represents the WB images in which endogenous control actin seems to be very low for 100 µg of protein as mentioned in the methods section.

We thank the reviewer for the opportunity to correct our mistake. In fact, we loaded 20 µg (as already indicated in the figure legend) of cell lysates and not 100 as wrongly stated. We have corrected it

  1. Reference missing for calpain activity measurement assay

We have inserted a reference for calpain activity measurement.

Minor comments

Some grammar and typo error throughout the MS some of these mentioned below

3,7% paraformaldehyde (PFA) should be 3.7 % paraformaldehyde (PFA)

Is it 5% crystal violet or 0.5 %?

We have correct the mistakes.

Reviewer 2 Report

The authors present evidence of a new target and agent for treating mesothelioma.  

1) The authors should acknowledge recent developments and revise the comments on current mesothelioma treatment to include the Checkmate 743 trial (Lancet. 2021 Jan 30;397(10272):375-386.) trial and regimen (ipi/nivo).  This immunotherapy approach is largely supplanting platinum based chemotherapy as first line therapy in North America and Europe.

2) In light of the Checkmate 743 results, a brief discussion of how BOLD100 could interact with immunotherapy would be warranted.

Author Response

The authors present evidence of a new target and agent for treating mesothelioma. 

  • The authors should acknowledge recent developments and revise the comments on current mesothelioma treatment to include the Checkmate 743 trial (Lancet. 2021 Jan 30;397(10272):375-386.) trial and regimen (ipi/nivo). This immunotherapy approach is largely supplanting platinum based chemotherapy as first line therapy in North America and Europe.

2) In light of the Checkmate 743 results, a brief discussion of how BOLD100 could interact with immunotherapy would be warranted.

We thank reviewer for the positive evaluation of our ms. We have inserted in the introduction a statement concerning the Checkmate 743 trial. We have also briefly discussed, in the discussion section, the interaction of BOLD-100 with immunotherapy.

Reviewer 3 Report

The authors dissected very well the BOLD-100 mechanism of action from a biological point of view. Their work deserves to be published, provided that some few mistakes will be corrected.

COMMENTS:

Results:

cytotoxic assay: the cell line MET5a is indicated as “ mesothelioma”. Please correct.

ROS alteration after BOLD-100 treatment: while tBHP can affect the production of ROS in different cells, in this context it is used as an aspecific oxidant for DHR. Therefore I think it is better to omit that part of the sentence which refers to it as a ROS inducer: ...........a postive control ROS inducer (Figure 4B).

Author Response

The authors dissected very well the BOLD-100 mechanism of action from a biological point of view. Their work deserves to be published, provided that some few mistakes will be corrected.

COMMENTS:

Results:

cytotoxic assay: the cell line MET5a is indicated as “ mesothelioma”. Please correct.

ROS alteration after BOLD-100 treatment: while tBHP can affect the production of ROS in different cells, in this context it is used as an aspecific oxidant for DHR. Therefore I think it is better to omit that part of the sentence which refers to it as a ROS inducer: ...........a positive control ROS inducer (Figure 4B).

We thank reviewer for the positive evaluation of our ms.

We have corrected the mistake in Results sections as well as we have modified statement about tBHP.

Reviewer 4 Report

In this manuscript, Ranzato and colleagues explored the efficacy of the compound BOLD-100 to activate cell death in malignant pleural mesothelioma (MPM) cell lines. BOLD-100 is a specific modulator of the chaperone GRP78, a protein that authors previously found over-expressed in MPM cells.

In this investigation authors want to demonstrate that BOLD-100 activates the ER stress response with consequent ROS production and increase of Ca2+ transmission from endoplasmic reticulum (ER) to mitochondria, and, finally, cell death.

However, in my opinion, the experimental models and the experiments performed by authors are not sufficient and adequate to confirm their hypothesis. Furthermore, the investigation lack to unveil a clear molecular mechanism or explore the efficacy of the proposed compound to kill MPM cells in a systemic model. Following, I report some of the numerous concerns that have determined my decision.

Regarding the cellular model, authors have performed experiments in two commercial MPM cell line (REN and MM89) and used as control Met5A cells, which are a non-cancerous human mesothelial cell line. I consider this the first great limit of the work: authors should use primary human MPM cells and compare them with primary pleural mesothelial cells. Furthermore, authors affirm that since epithelioid MPM isotype is the most common, they decide to perform all experiment in REN cell only. I think this is not a correct explanation and they had to use diverse MPM cell type.

Regarding the experiments, the first set of experiments are not sufficient to prove the effects of BOLD-100 to increase the cell death program. In Figure 1 there are only graphs: where are the representative figures? Furthermore, they should also investigate about cell proliferation, migration and invasion. Similarly, I recommend authors to perform apoptosis experiments by also using immunoblot or FACS analysis and give representative illustrations of the apoptotic activity and include measurements of necrosis. Overall, a lot of graphs are enclosed throughout the manuscript but not the original experiments in which they obtained the data.

Next, authors want to demonstrate that by using an inhibitor of GRP78 is possible to activate ER-stress and provoke increase of the ER-Calcium release necessary to activate the mitochondrial damage and finally apoptosis. Where are the experiments demonstrating that the mitochondrial and ER compartment are affected? Next, I consider the calcium measurements not adequate because the only detection of intracellular calcium is not sufficient. They had to detect the mitochondrial calcium levels and ER-calcium release and also verify that there is no effect on the plasma membrane channels/pumps. GRP78 is determinant to preserve the juxtaposition between ER and mitochondria. Why have authors not verified these ER-mitochondria associations? In addition, authors should also perform experiments by silencing and overexpressing GRP78 to exclude possible side effects of the compounds used. Similarly, caffeine is not the best method to block IP3Rs activity. In line with this, which isoform are they inhibiting? Finally, UPR activation cannot detect by only mRNA quantification of CHOP and XBP1. Further experiments should be performed.

In conclusion, I consider this manuscript not suitable for Cancers

Author Response

In this manuscript, Ranzato and colleagues explored the efficacy of the compound BOLD-100 to activate cell death in malignant pleural mesothelioma (MPM) cell lines. BOLD-100 is a specific modulator of the chaperone GRP78, a protein that authors previously found over-expressed in MPM cells.

In this investigation authors want to demonstrate that BOLD-100 activates the ER stress response with consequent ROS production and increase of Ca2+ transmission from endoplasmic reticulum (ER) to mitochondria, and, finally, cell death.

However, in my opinion, the experimental models and the experiments performed by authors are not sufficient and adequate to confirm their hypothesis. Furthermore, the investigation lack to unveil a clear molecular mechanism or explore the efficacy of the proposed compound to kill MPM cells in a systemic model. Following, I report some of the numerous concerns that have determined my decision.

Regarding the cellular model, authors have performed experiments in two commercial MPM cell line (REN and MM89) and used as control Met5A cells, which are a non-cancerous human mesothelial cell line. I consider this the first great limit of the work: authors should use primary human MPM cells and compare them with primary pleural mesothelial cells. Furthermore, authors affirm that since epithelioid MPM isotype is the most common, they decide to perform all experiment in REN cell only. I think this is not a correct explanation and they had to use diverse MPM cell type.

We thank reviewer for the analysis of our work. We know that MeT5A cells are not a primary mesothelioma cell line, but we and other authors use them as a starting model for comparison with mesothelioma cells. To strengthen our study, we compared cytotoxicity also in mesothelial cells derived from pleural exudates. In this way, we confirmed the data and the behavior of the MeT5A, which are therefore a good model.

Moreover, we decided to not use primary mesothelioma cells because they are not well characterized and not shared by other laboratories. So, we have obtained data on other already worldwide available mesothelioma cells lines to further corroborate our observations. And the behavior of BOLD-100 on Ca2+ variations is similar on MM98 and MSTO211H cell lines, suggesting a shared mechanism (supplementary figure 1 for review process).

Regarding the experiments, the first set of experiments are not sufficient to prove the effects of BOLD-100 to increase the cell death program. In Figure 1 there are only graphs: where are the representative figures? Furthermore, they should also investigate about cell proliferation, migration and invasion. Similarly, I recommend authors to perform apoptosis experiments by also using immunoblot or FACS analysis and give representative illustrations of the apoptotic activity and include measurements of necrosis. Overall, a lot of graphs are enclosed throughout the manuscript but not the original experiments in which they obtained the data.

We have inserted in Figure 1 representative figures of colony formation assay, as well as we performed migration assay that showed similar behavior for REN, MM98 and MSTO211H cell lines (supplementary figure 2 for review process), demonstrating an inhibition of migrating behavior in presence of BOLD-100 treatment.

Moreover, the determination of caspase activation can be performed in different ways. The detection of activated caspases could be performed by immunoblotting. This is a method by which the processing of a particular pro-caspase is studied by detecting (with a specific antibody) the appearance of small or large subunit of the mature caspase. The main limitation of the immunoblot technique is the requirement for high quality antibodies reacting specifically with individual caspases, as well as immunoblotting is mainly a qualitative assay (Kohler et al., Journal of Immunological Methods 2002).

One of the commonly used ways to detect caspase activity is to measure the cleavage of synthetic substrates upon their incubation with lysates of apoptotic cells. The substrate is cleaved by caspases recognizing the substrate cleavage site, thereby giving rise to increased color or fluorescence intensity.

The intensity of the signal is proportional to the amount of cleaved substrate, which in turn, is dependent on caspase activity and is related to the percentage of apoptotic cells in a cell population.

Detection of caspase activity by cleavage of synthetic substrates is a commonly used method and has several advantages. It is a quantitative, convenient, relatively fast and sensitive method.

So, we decided to study the apoptotic pathway involvement in BOLD-100 toxicity in our cell model using fluorescent synthetic substrates.

In order to avoid redundancy in the sequence specificity of caspases for substrates, i.e. many caspases have the ability to cleave the same substrates, at least in vitro, although with different efficiency, it is possible to use, as we done, specific caspase inhibitors. In particular, we used a caspase 3/7 inhibitor (N-Ac-DEVD-CHO) and a specific inhibitor of caspase-8 (Z-IETD-FMK).

Next, authors want to demonstrate that by using an inhibitor of GRP78 is possible to activate ER-stress and provoke increase of the ER-Calcium release necessary to activate the mitochondrial damage and finally apoptosis. Where are the experiments demonstrating that the mitochondrial and ER compartment are affected? Next, I consider the calcium measurements not adequate because the only detection of intracellular calcium is not sufficient. They had to detect the mitochondrial calcium levels and ER-calcium release and also verify that there is no effect on the plasma membrane channels/pumps. GRP78 is determinant to preserve the juxtaposition between ER and mitochondria. Why have authors not verified these ER-mitochondria associations? In addition, authors should also perform experiments by silencing and overexpressing GRP78 to exclude possible side effects of the compounds used. Similarly, caffeine is not the best method to block IP3Rs activity. In line with this, which isoform are they inhibiting? Finally, UPR activation cannot detect by only mRNA quantification of CHOP and XBP1. Further experiments should be performed.

In conclusion, I consider this manuscript not suitable for Cancers

The reviewer is absolutely right. The interaction between ER and mitochondria and the role of GRP78 is absolutely of importance in understanding the role of GRP78 in mesothelioma and even more the effect of BOLD-100 in this interaction. We have not evaluated this aspect as it is very complicated, but we are working on another manuscript with mitochondrial Ca2+ experts to explore this aspect.

High concentrations of caffeine (10–70 mM) inhibited Ca2+ release via IP3R1 (Huma Saleem et al., Interactions of antagonists with subtypes of inositol 1,4,5-trisphosphate(IP3) receptor, British Journal of Pharmacology (2014) 171 3298–331).

Interestingly novel mechanistic insights in the regulation of ER Ca2+ homeostasis by GRP78 showed as IP3R1 knock down induced an increase of susceptibility of cells towards ER-stress inducers, leading to a decrease in mitochondrial potential and increased apoptosis (Higo et al., Mechanism of ER stress-induced brain damage by IP(3) receptor, Neuron 2010).

Conversely, cells treated with ER-stress inducers significantly impaired IP3R-channel activity in intact cells and microsomal preparations. The molecular mechanisms involved a direct binding of GRP78 to IP3R1. In fact, GRP78 selectively bound to IP3R1, but did not bind to the other IP3R isoforms. More important, during ER stress conditions, GRP78 dissociates form IP3R1. This decrease in GRP78 binding to IP3R1 during ER stress seems to underlie the impaired IP3R1-channel activity, since GRP78 knock down impaired Ca2+ release through IP3R1, not through IP3R2 or IP3R3. This indicated that GRP78 binding to IP3R1 is essential for IP3R1-channel activity (Agostinins, Samali (Eds), Endoplasmic reticulum stress in health and disease, Springer 2012).

Moreover, we believe that the presented data allow us to define and outline a mechanism of action of BOLD-100 in mesothelioma. Indeed, we believe we have outlined an interesting mechanism of action, never before described for the BOLD-100 that allows us to explore its effect alone or in synergy in mesothelioma in the near future, representing a concrete possibility of curing this neoplasm.

We have already demonstrated the feasibility of UPR activation by qPCR (Martinotti et al., Journal of Cellular Physiology 2018). Moreover, data from Figure 7B derived from WB experiments.

In addition, the GRP78 silencing data show that the effect of BOLD-100 is much less pronounced (EC50 siRNA 424 µM (168-1074) vs EC50 Scrambled 91 µM (42-194)).

Round 2

Reviewer 1 Report

Authors have made necessary changes required in the manuscript and justified the concerns raised in the previous version of the manuscript and henceforth, I recommend the manuscript for the acceptance. 

Reviewer 2 Report

The authors have adequately addressed the reviewer's comments.

Reviewer 4 Report

Dear Authors, 

thanks to have improved the manuscript by using other cellular samples, including primary mesothelial cells and other mesothelioma cell lines. 

However, apart this new set of experiments you did not perform any experiments that really account  that the investigated compounds can modulate the apoptosis by regulating the intracellular calcium dynamics, in particular the ER-mitochondrial calcium transfer.  

In this version, the manuscript still lack of solid data and a correct experimental approach.

Finally, I agree with you that that caffeine has been already used to modulate the IP3Rs activity. (Huma Saleem et al., Interactions of antagonists with subtypes of inositol 1,4,5-trisphosphate(IP3) receptor, British Journal of Pharmacology (2014) 171 3298–331). However, experiments conducted by this research group were achieved in permeabilized IP3Rs-ko cells stable expressing the different isoforms of IP3Rs.